# Experimental and Numerical Study of Al2219 Powders Deposition on Al2219-T6 Substrate by Cold Spray: Effects of Spray Angle, Traverse Speed, and Standoff Distance

**DOI:** 10.3390/ma16155240

**Published:** 2023-07-26

**Authors:** Zheng Zhang, Tzee Luai Meng, Coryl Jing Jun Lee, Fengxia Wei, Te Ba, Zhi-Qian Zhang, Jisheng Pan

**Affiliations:** 1Institute of Materials Research and Engineering (IMRE), Agency for Science, Technology and Research (A*STAR), 2 Fusionopolis Way, Innovis #08-03, Singapore 138634, Singapore; meng_tzee_luai@imre.a-star.edu.sg (T.L.M.); coryl-lee@imre.a-star.edu.sg (C.J.J.L.); wei_fengxia@imre.a-star.edu.sg (F.W.); js-pan@imre.a-star.edu.sg (J.P.); 2Institute of High Performance Computing (IHPC), Agency for Science, Technology and Research (A*STAR), 1 Fusionopolis Way, Connexis #16-16, Singapore 138632, Singapore; zhangz@ihpc.a-star.edu.sg

**Keywords:** cold spray, spray angle, traverse speed, standoff distance, finite element modeling, CFD

## Abstract

Cold spray (CS) is an emerging technology for repairing and 3D additive manufacturing of a variety of metallic components using deformable metal powders. In CS deposition, gas type, gas pressure, gas temperature, and powder feed rate are the four key process parameters that have been intensively studied. Spray angle, spray gun traverse speed, and standoff distance (SoD) are the other three process parameters that have been less investigated but are also important, especially when depositing on uneven substrates or building up 3D freeform structures. Herein, the effects of spray angle, traverse speed, and SoD during CS deposition have been investigated holistically on a single material system (i.e., Al2219 powders on Al2219-T6 substrate). The coatings’ mass gain, thickness, porosity, and residual stress have been characterized, and the results show that spray angle and traverse speed exercise much more effects than SoD in determining coatings’ buildup. Finite element method (FEM) modeling and computational fluid dynamic (CFD) simulation have been carried out to understand the effects of these three parameters for implementing CS as repairing and additive manufacturing using aluminum-based alloy powders.

## 1. Introduction

Cold spray (CS) is the latest member of the thermal spray family. Unlike other thermal spray techniques, such as atmospheric plasma spray (APS) or high velocity oxyfuel spray (HVOF), which mainly use thermal energy to melt or soften powders, CS dominantly utilizes powders’ kinetic energy to soften powders only (without melting) before depositing them into coatings or 3D freeform structure [1]. In CS, the metallic powders (1–60 μm) can be propelled by air or inert gas (helium, nitrogen, or their mixture) to a supersonic speed (300~1200 m/s) after compressing the gas to a pressure of 2~8 MPa with or without heating the gas (maximum up to 1100 °C). Once powders’ velocity is above a critical value (i.e., 620~660 m/s [1,2] for Al), their significant kinetic energy results in severe plastic deformation of powders and an increase of temperature at the contact zone. Both effects help remove surface oxides at the contact point for both powder and substrate. A fresh metallic bonding is thus created at the interface. Micrometer thick coatings and 3D freeform structures, such as aluminum alloy tube and flange [3], aluminum, and copper stiffeners [4] can be obtained without significant oxidation and phase transformation.

Thick metallic coatings have been successfully deposited by cold spray using various metallic powders, including aluminum (Al), copper (Cu), nickel (Ni), commercial pure titanium (Ti), silver (Ag), zinc (Zn), chromium (Cr), as well as compound alloy powders such as Ti-6Al-4V [5,6], stainless steel, nickel-based alloy [7,8] and CoNiCrAlY powders [3,9], etc. Metal powders which have a large number of slips systems, such as face-center-cubic (FCC) metal (including Al, Cu, Ni, and 316 stainless steel), are observed to exhibit high deformability with low porosity upon CS deposition. Through these studies, the effects of key process parameters, including gas type, gas temperature, gas pressure, and powder feed rate have been extensively investigated, as they affect the powder impact velocity significantly [9]. The in-depth knowledge of such process parameters has led to the adoption of low-pressure cold spray system to deposit Al6061 powders for non-structural applications in Honeywell Aerospace, such as dimension restoration and repairing for fretting, pitting, corrosion, and seal surface, etc. [10].

Besides the above-mentioned parameters, there are other process parameters, including spray angle, spray gun traverse speed, and standoff distance (SoD), which are less investigated but also exercise significant influences in the final deposits based on the limited investigations thus far. The studies about the effects of spray angle when depositing pure Al [11], commercial pure Ti (CP Ti) [12,13,14], Ti-6Al-4V [5,6], Cu [12,15], Ni [16], and Inconel 718 [8,17] powders reveal that there exists a minimum spray angle for deposition to take place. This minimum spray angle is also known as the critical angle, which depends on both powder and substrate materials. The critical angle (with respect to the horizontal plane) was 50° for depositing pure aluminum powders on Al2024-T3 coupons [11], while it was 45° for depositing CP Ti powders on AlMg_3_ substrate [13]. An initial increase from the critical angle usually leads to an increase in deposition efficiency with a concurrent reduction in the coating’s porosity. However, different materials have different trends of behavior in terms of coating thickness and mass buildup when the spray angle increases from the critical angle up to 90°. Our recent studies show that coating thickness and mass buildup have a turning point at 70° spray angle when depositing hexagonal-close-packed (HCP) Ti-6Al-4V powders on Al6061-T6 substrates [5,6], while they have a monotonic relationship with spray angle up to 90° when depositing face-center-cubic (FCC) Inconel 718 powders on the same Al6061-T6 substrate [8]. However, another FCC metal, pure Al powders [11], has been reported to have an increase in its thickness and relative deposition efficiency with a spray angle from 50 up to 70° before it saturated at 80 and 90°. 

Traverse (or transverse) speed has a direct influence on the deposit’s thickness per pass and the buildup of heat at the substrate. A lower transverse speed usually leads to higher substrate temperature, a thicker layer per pass, an increased deposition efficiency [18,19,20,21], and a lower porosity when depositing Ti64 [19], CoCr powders [20], and Cu powders [21]. 

The standoff distance (SoD) affects the particle velocity and the resultant deposition efficiency through the complex interplay between particle acceleration/deceleration by the gas jet and particle deceleration by the bow shock [22,23,24]. Bow shock forms because of the supersonic gas flow’s adjustment upon hitting the substrate downstream of the gas flow, which leads to the formation of curved and detached shockwaves at the substrate [23]. If SoD is too small (i.e., <2 cm), a bow shock can form in between the nozzle and the substrate, and the bow shock is detrimental to the deposition process as it reduces particle impact velocity [23]. If SoD is too large (>10 cm), powder velocities are higher than the gas velocity and the negative drag force decelerates the powder undesirably. Using air as propellant gas with SoD ranging between 1 and 11 cm, Li et al. observed that the relative deposition efficiencies (DE) of Al2319 and CP Ti coatings decreased with an increase in SoD, while relative DE of pure Cu coating increased from 1 to 3 cm before the Cu coating decreased with SoD beyond 3 cm [22]. Shorter SoD also leads to faster heat buildup in coating and substrate [23,24], which may lead to easier oxidation of the coatings if using air as propellant gas. Although the optimum SoDs typically range from 2 to 6 cm, they vary among different materials and different systems. SoDs as large as 7 cm and 8 cm have been reported for CP Ti powders (on Al7075 substrate) [25] and IN625 powders (on IN625 substrate) [26], respectively. 

The above investigations show that spray angle, traverse speed, and SoD influence the cold spray deposition significantly and that they behave differently among different powder–substrate combinations. There is a lack of research on these three parameters holistically on a single material system to evaluate their relative effectiveness. There are no reports on the effects of spray angles and traverse speed on Al alloy powders. Therefore, in this work, we probe the effects of these three parameters systematically on one material system (Al2219 powders on Al2219-T6 substrates) by using three specially designed fixtures. Aluminum 2219 alloy uses copper (Cu) as the principal alloying element and the Al_2_Cu precipitate as the hardening phase. Al2219 has been widely used for high-temperature structural applications, including aircraft skin, space boosters, and fuel tanks in the aerospace industry due to its high yield strength, high fracture toughness, weldability, and resistance to stress corrosion cracking [27]. Using Al2219—Al2219-T6 as a common material system due to their technological relevance, the influence of spray angle, traverse speed, and SoD on coatings’ mass, thickness, porosity, and residual stress will be investigated experimentally. In addition, finite element method (FEM) modeling based on commercial software package Abaqus/Explicit in Eulerian framework (version 2018) will be carried out to determine the critical angle and to model the residual stress caused by different impact angles, while the computational fluid dynamic (CFD) simulation will be employed to study the effects of these three parameters. As a result, the influence of these three spray parameters can be better understood for deploying CS in repairing aluminum components and for additive manufacturing of aluminum-based 3D freeform structures, such as aluminum flanges [3], stiffeners [4], and other more complex structures.

## 2. Materials and Methods

Al2219 powders were purchased from TLS Technik GmbH and Co. (Bitterfeld-Wolfen, Germany). The surface morphology and chemical composition of the powders were examined under JEOL (Tokyo, Japan) JSM-7600F field emission scanning electron microscope (FE-SEM) equipped with X-Max 50 energy dispersive spectroscopy (EDS) detector (Oxford, UK). After dispersing the powders in deionized water using an ultrasonic bath, their size distribution was measured using Malvern Mastersizer 2000 particle size analyzer (Malvern Panalytical Ltd., Malvern, UK). One gram of the Al2219 was cold pressed into a pellet and mounted into an epoxy mold. After the entire resin was polished using standard metallurgical sample preparation to expose the cross-section of the powders, Agilent G200 nano-indenter (Keysight, CA, USA) was used to measure hardness and Young’s modulus of 30 powders with a Berkovich diamond tip using continuous stiffness measurement (CSM) mode down to 1000 nm indentation depth. JEOL IT500HR FE-SEM equipped with Oxford Instruments Symmetry EBSD (Oxford, UK) detector was used to analyze grain orientations of powders from their cross-section using 20 kV electron accelerating voltage and a scanning step size of 0.2 µm. Al2219 powders’ crystallinity was examined using Bruker D8 Advance X-ray diffraction (XRD, Freiberg, Germany) with Cu Kα X-ray (λ = 1.54 Å) at a voltage of 40 kV and a current of 40 mA. Rietveld phase analysis was further employed to quantify the phase composition using TOPAS v5 software and deploying a full axial model and fundamental parameters approach.

To holistically investigate the effect of spray angle, traverse speed, and standoff distance (SoD) on one material system, three stainless steel fixtures were fabricated after modifying two previous designs [11,12]. Each fixture has seven facets with a 90° to 30° angle to the normal direction (represented by a black dotted arrow) of the fixture’s bottom plane (Figure 1a), while the center of each facet has the same distance from the bottom plane. The third fixture (F3) is 3 cm higher than the second fixture (F2), which is 3 cm higher than the first fixture (F1). Al2219-T6 coupons were cut into the size of 3 × 1 × 0.3 cm and fixed onto each facet using two stainless steel screws at the two ends. Impact Innovation 5/11 cold spray system was used for deposition with nitrogen (N_2_) as the propellant gas, which was heated to 400 °C with a pressure of 6.0 MPa. The three fixtures were laid flat on the turning table (Figure 1b) with their bottom planes aligned (represented by the black dotted arrow) during the actual spray. The spray gun was positioned at the center of the fixture and parallel to the table surface so that the seven facets in each fixture had the same SoD (3 cm) between the spray nozzle exit and the third fixture (F3). Five passes were swiped back and forth across the center of three fixtures to study the effects of spray angle and SoD at the three traverse speeds, which varied from 200 (T1), 350 (T2) to 500 (T3) mm/s. These three values represent low, medium, and high traverse speeds during cold spray deposition. The spray matrix can be found in Table 1 below.

Each coupon’s weight change was measured by a weighing balance with an accuracy of 0.1 mg before and after deposition so that the weight change at each condition can be calculated accurately. The deposits’ vertical profiles were analyzed by KLA Tencor P-10 surface profilometer (Newport Beach, CA, USA). Surface residual stress of the deposits was evaluated with chromium (Cr) Kα X-ray and vanadium (V) filter using Stresstech Xstress G3 X-ray stress (Jyväskylä, Finland) measurement by detecting Al (311) diffraction plane at 139.3°. X-ray spot was focused on the center of each coupon where the deposits (if any) were thick. After these non-destructive measurements, a slice of the deposit along the longitudinal direction of the coupon was cut for surface morphology and cross-section microstructure analysis. The deposits’ surface morphology was also examined under JEOL JSM-7600F SEM. After grinding and polishing to expose the cross-sections of the deposits, they were examined under Olympus BX53M optical microscope, and the captured images were further analyzed using ImageJ software (version 1.50i) for coating’s thickness and porosity, which are only reported for continuous Al2219 deposits (but not for those forming discontinuous coatings). Deposits’ cross-sections were further examined by SEM (under backscattering mode) and EBSD to reveal the powders’ deformation and microstructure after deposition with a step size of 0.1 μm. Nano-indentation was similarly carried out to analyze deposits’ hardness and Young’s modulus at their cross-sections.

The 3D finite element model (FEM) was developed using Abaqus/Explict in the Eulerian framework to simulate the single particle impact behaviors during the coating process [5,28,29]. The single particle impact model with the particle diameter of 43.8 µm (which is the D_50_ measured by the experiment) was used to study the critical angle for bonding, while the multiple particle impact model was employed to study the residual stress generated during the coating processes. Detailed information about single and multiple particle impact FEM simulation models is provided in Section 3 and Section 4 of the Appendix A. The material of the powder particles and substrate is modeled as Johnson–Cook (JC) plasticity model in Equation (1) which can address the material behaviors influenced by strain hardening, strain rate hardening, and thermal softening. The Al2219 material parameters are listed in Table 2.
(1)$$\sigma =\left(A+B\varepsilon^{n}\right)\left[1+Cln\left(1+\frac{\dot{\epsilon }}{\dot{\varepsilon_{0}}}\right)\right]\left(1-{\left[\frac{T-T_{room}}{T_{melt}-T_{room}}\right]}^{m}\right)$$

Navier–Stokes equation was solved in the computational fluid dynamic (CFD) simulation, with the computational domain, including the chamber, the nozzle, the powder injector, and a large domain surrounding the nozzle exit. The nozzle was a De Laval nozzle with an outlet diameter of 6.4 mm, an expansion ratio of approximately 5.8, and a divergent part of 130 mm long. The real gas formula was used to model the energy state of nitrogen, while the same gas pressure and temperature as the actual spray experiment, were applied to the gas and powder inlet of the cold spray setup. After a steady free stream flow field was obtained, powders were injected into the system at the powder inlet, which exactly followed the size distribution of the powder feedstock. The discrete phase model (DPM) was used to calculate the particle dynamics via calculating the interactions between the gas flow and the powders (discreate phase). The interaction between the powders was also taken into consideration.

## 3. Results

SEM image shows that the as-received Al2219 powders dominantly had a spherical shape (Figure 2a). The majority of the powders had a diameter of about 42.5 μm. There were occasional satellite particles (less than 5 μm) attached to the large powders (indicated by rectangular box) and agglomeration of small powders with a size of less than 15 μm (indicated by the yellow ellipses). Chemical composition by EDS showed that the Al2219 powders were comprised of 92.8 wt% of Al, 6.1 wt% of Cu, 0.4 wt% of Mn, 0.3 wt% of Fe, 0.2 wt% Zr, 0.1 wt% of Si, and 0.1 wt% of V. The particle size distribution (PSD) analysis determined D_10_, D_50_, and D_90_ to be 31.9, 43.8, and 59.4 μm, respectively (Figure 2b). There was also a small hump around 18 μm in PSD, which corresponds to the agglomeration of fine powders, as indicated by yellow ellipses in Figure 2a.

A dendritic structure can be observed inside the cross-section of each powder (Figure 2c). Elemental mapping of copper in the right corner of Figure 2c shows that copper as the main alloying element was preferentially distributed in the dendritic regions as the strengthening grain boundaries. Rietveld phase analysis of Al2219’s XRD pattern shows that the powders were comprised of 93.46 wt% of Al phase and 6.54 wt% Al_2_Cu phase (Figure 2d). The inverse pole figure in z direction (Figure 2e) from electron beam backscattering (EBSD) shows that each powder consists of randomly distributed grains with sizes ranging from 0.7 to 15.6 μm. The phase map at the top right corner demonstrates that the powder was dominated by face-center-cube Al phase (red color) with Al_2_Cu (blue color) mainly distributed at the grain boundaries (Figure 2e). The hardness and Young’s modulus of the Al2219 powders were taken to be 1.41 GPa and 68.09 GPa, respectively, from the averaged value between 800 and 900 nm (marked by blue dotted rectangle box) in the nano-indentation test (Figure 2f).

The entire pictures from nine sets of Al2219 coupons deposited at three traverse speeds and three standoff distances (SoD) with seven spray angles can be seen in Appendix A. Two of nine sets after depositing using the lowest traverse speed (200 mm/s, T1) with the smallest SoD (3 cm, F3) as well as the fastest traverse speed (500 mm/s, T3) with the largest SoD (9 cm, F1) were selected and shown in Figure 3. Institutively, the former (T1F3) and the latter (T3F1) sets were expected to have the highest and lowest deposition of Al2219 powders, respectively. In both sets, the 90° spray angle had the thinnest width of the deposits among the seven angles, which gradually increased when the spray angle decreased from 90 to 50°. At the same spray angle (i.e., 90°), the deposit’s width was narrower at the shorter SoD (3 cm, T1F3, Figure 3a) than at the larger SoD (9 cm, T3F1, Figure 3b). It was visually difficult to determine whether there were deposits or just erosion marks at 40 and 30° spray angles, which can be seen much clearer from the deposits’ vertical profile under the surface profiler (Figure 3a′,b′). At low traverse speed (T1, 200 mm/s), both spray angles at 90 and 80° led to similar deposits’ heights which peak at about 120 µm, represented by black and red curves in Figure 3a′. At fast traverse speed (T3, 500 mm/s), 90° spray angle led to a higher deposit’s peak height of 70 µm compared to that of the deposits (60 µm) sprayed at 80°, represented by black and red curves in Figure 3b′. At both traverse speeds, the deposits’ heights were clearly lower when sprayed at 70° (green curves) than those sprayed at 80 and 90°. There were only a few spikes at 60° and purely consisted of background at spray angles from 50 to 30°, which indicates no deposits at spray angles from 50 to 30°. 

Besides measuring surface profiles of the nine sets of coupons, other non-destructive measurements, including weight change and surface residual stress, were also performed. Thereafter, a small piece of deposit shown by the dotted box in Figure 3a was cut from each coupon and was subjected to SEM and EDS examination of the surface morphology and composition, followed by cross-sectional examination under optical microscope, SEM, and EBSD. The surface morphology of T1F3 and T3F1 sets of samples are shown in Figure 4, while the SEM images from the other seven sets of coupons are presented in Appendix A in Supplementary Information. For the T1F3 set of samples, the number of spherical powders with a D_50_ diameter (around 43.8 μm) clearly decreased when the spray angle reduced from 90 to 70° at the current picture frame, with only two powders at 60° and zero powders at 50, 40, and 30° (Figure 4a–g). Instead, more craters appeared at a 60° spray angle by the un-bonded powders, which impacted the surface and bounced away. Similar craters dominated the entire surface from 50 to 30°. The craters’ depth was deeper at 60° compared to that at 30°. It thus indicates a decrease in powder buildup and deposition efficiency (DE) when the spray angle decreased from 90 to 70°, with negligible deposition when the spray angle was 60° and below. The same trend of decreasing DE with spray angle was observed in the T3F1 case (Figure 4a′–g′), except that the number of D_50_ powders was already low at 80° spray angle and was negligible when the spray angle was at 70° and below. At the same spray angle, the craters’ depth was deeper at smaller SoD (3 cm, T1F3) compared to that at larger SoD (9 cm, T3F1). T1F1 and T1F2 sets of samples have the same traverse speed as T1F3, and their surface morphologies at seven spray angles (Appendix A) are similar to those of T1F3 in (Figure 4a–g). In contrast, by increasing the traverse speed from 200 (T1) to 500 m/s (T3), the Al2219 buildup has been reduced significantly. Hence, traverse speed exercises a much more significant effect than standoff distance on the buildup rate.

At the slow transverse speed (200 mm/s, T1) and small SoD (3 cm, F3), cross-sections of the Al2219 deposits show continuous coatings at three high spray angles of 90, 80 and 70° (Figure 5a–c and Figure 6a–c) with intimate interfaces but with decreasing thickness clearly. Under SEM, the Al_2_Cu precipitates within initial Al2219 powders (Figure 2c) were still preserved as bright grain boundaries in all Al2219 deposits (Figure 6), while Al_2_Cu precipitates in Al2219-T6 substrate also appeared as bright areas (highlighted by ellipses) due to enhanced backscattering electrons from the Cu atoms which are heavier than Al atoms. At a 60° spray angle, only single powder was deposited occasionally (Figure 6d), with dented surfaces in other areas. No powders were deposited from 50° to 30°, and only dented surface peened by unbound Al2219 powders can be seen from the cross-sections (Figure 5d–g).

At the fast transverse speed (500 mm/s, T3) with large SoD (9 cm, F1), even 90° was unable to produce a continuous coating (Figure 5a′ and Figure 6a′). Only scattered deposits with very thin thickness (~37 µm) after five passes can be observed. Such thickness was equivalent to a single-layer deposition. The deposits at 80 and 70° spray angles (Figure 5b′,c′ and Figure 6b′,c′) were also discontinuous and thin. Single powder was similarly observed to deposit at 60° spray angle (Figure 5d and Figure 6d′), while there were only dented surfaces due to crater formation for 50 to 30° spray angles (Figure 5d′–g′ and Figure 6e′). These results match the surface profiler’s results in Figure 3a′,b′ well.

The grain orientation and deformation of the Al2219 powders after deposition can be seen via EBSD in Figure 7 for spray angles of 90 and 70° in the T1F3 set of samples. Despite using a smaller step size of 0.1 μm than 0.2 μm which was used for analyzing the initial Al2219 powders in Figure 2e, a large portion of Al2219 deposits at both 90 and 70° as well as the top substrates had very low indexing, represented by the black colors in IPF map of coating areas and top of the substrates (Figure 7b,d). Such a low indexing rate indicates that the supersonic impact has severely fragmented Al2219 powders into very fine grains, which are smaller than the EBSD’s step size of 0.1 µm. Other cold-spray-deposited Al6061 [29], Inconel 718 [7,8], and Ti64 [5,6] have been observed to have similar fragmentation and low indexing rates. For the Al2219 powders, which have not been fragmented severely, such as the top of Al2219 deposits at 70° (marked by a rounded rectangle), they clearly show much better indexing than the center coating (marked by a rectangle in Figure 7d).

After analyzing all 63 coupons with the combination of seven spray angles, three traverse speeds, and three SoD, the resultant deposits’ weight change, coating thickness, porosity, and residual stress are shown in Figure 8. The weight increase (Figure 8a) shows that spray angle and traverse speed are the two process parameters that determine the coating buildup most significantly. The minimum angle where noticeable weight gain can only be observed is 60° (T1F3, the green upward triangle in Figure 8a), indicating that the critical angle for depositing Al2219 on Al2219-T6 substrate is about 60°. No deposition but erosion took place below the 60° spray angle. A decrease in traverse speed leads to an increase in weight gain at angles larger than 70°. No weight gain at angles below 60° in Figure 8a agrees well with only craters observed in the same samples under SEM (Figure 4 and Figure 6) and the optical microscope (Figure 5).

In the three spray angles (70, 80, and 90°) where there were deposits, continuous coatings can only be observed at the nine samples deposited with a low traverse speed of 200 mm/s (T1) as well as six samples deposited with a medium transverse speed of 350 mm/s (T2). Al2219 deposits’ thickness and porosities were measured only at these 15 samples and are shown in Figure 8b,c. Low traverse speed led to a thicker coating at the same spray angle (Figure 8b). At low traverse speed (T1, 200 mm/s) and large SoD (F1, 9 mm), a 70° spray angle only led to a thickness of 53.6 ± 16.0 µm, corresponding to 1~2 layers of powders after shear deformation (Figure 8b). It thus indicates very low deposition efficiency at low spray angles. At the same traverse speed, reducing the spray angle led to thinner coating (Figure 8b), while porosities in these continuous coatings generally decreased with an increase in spray angle (Figure 8c). An increase in transverse speed and SoD led to higher porosity, as seen in the T2F1 and T2F2 series of samples (Figure 8c).

All 63 coupons had compressive residual stress (CRS) at the surface, represented by the negative residual stress values in Figure 8d. For the 15 continuous coatings deposited at spray angles from 70 to 90°, the CRS was in the range between −24 and −80 MPa (represented by solid symbols in Figure 8d). There was an increase in CRS value with a decrease in spray angles from 90 to 70°. Both the CRS magnitude and the decreasing trend of CRS with spray angle were similarly observed in Al6061 coatings deposited by CS on Al6061-T6 substrate at three spray angles of 50°, 70° and 90° [28]. The CRS increased to around −200 MPa when the spray angle decreased to 60~40° with a slight recovery at 30°. These unusually high CRS (represented by open symbols in Figure 8d) was attributed to the shot-peened Al2219-T6 substrates by the unbonded Al2219 powders and did not reflect the CRS from the discontinuous Al2219 deposits.

The hardness and Young’s modulus of Al2219 deposits which formed continuous coatings were examined under nano-indentation test from their cross-sections. It can be seen that the Al2219 coatings’ hardness and Young’s modulus were around 1.55~1.73 GPa and 71.13~81.50 GPa, respectively (Table 3). Both values were higher than hardness and Young’s modulus of 1.41 GPa and 68.09 GPa from the initial Al2219 powders (Figure 2f), which was attributed to the work-hardening of the deposits caused by grain refinement after the high impact velocity. There was no visible difference in the hardness and Young’s modulus of the coatings deposited at different traverse speeds, spray angles, and standoff distances. The similarity in hardness values between two different spray angles (60° and 90°) was also observed in Inconel 718 coatings deposited on Inconel 718 substrate by CS [17].

## 4. Discussion

The weight change of the Al2219 deposits (Figure 8a) clearly demonstrates that spray angle and traverse speed affect the CS buildup rate much more significantly than standoff distance (SoD). Spray angle has exercised a similarly significant effect when depositing pure Al powders on Al2024 substrates [9] and Al6061 powders on Al6061-T6 substrate [27]. The soft Al powders had a wide spray angle window at 70°~90°, which resulted in a high deposition rate, before a transient region at 50°~60° with a medium deposition rate followed by a no-deposition region at 30°~40° [9]. Given that both Al6061 and Al2219 powders are harder than pure Al powders due to their alloying elements, it is not surprising to see that both Al6061 [27] and Al2219 (Figure 8a) have a much narrower spray angle range at 80°~90° for a high deposition rate followed by a narrow transient region at 60°~70° with medium deposition rate. While there were low deposition rates at 40°~50° for Al6061, no deposition occurred when the impact angle was 50° and below for Al2219. Hence, the critical angle for building up Al2219 coating on Al2219-T6 substrate is 60°. Finite element method (FEM) modeling was carried out by simulating single Al2219 powder (43.8 μm) impacting Al2219-T6 substrate from an angle of 70 to 45°. The resulting material velocity profiles after 1 ms can be seen in Figure 9, which shows that the Al2219 powder was able to stick to the substrate when the impact angle decreased from 70 to 65 and to 60°. A further decrease in impact angle to 55° and below led to slipping the entire powder from the surface. The FEM simulation agrees with the critical angle (60°) observed in the experiments (Figure 8a).

Besides studying the critical angle for bonding, FEM model is also employed to simulate multiple powders’ impact and the subsequent development of residual stress at different impact angles (Figure 10). The diameters of the powders in the FEM model (Figure 10a) follow the measured powder size distribution in Figure 2b. Several layers of the powders are deposited in the simulation to form about 60 µm thick coatings with the assumption that all powders are deposited. The residual stresses are investigated after a sufficiently long simulation time so that the system’s kinetic energy is almost entirely dissipated. Figure 10b displays an example of simulation results of equivalent plastic strain (PEEQ) and von Mises stress after the deposition in 90° impact angle. Figure 10c illustrates the residual stress distribution through the whole thickness of the coating achieved by FEM simulation with different spray angles, where x = 0 indicates the coating-substrate interface. The results demonstrate that the in-plane residual stress in the whole coating along the nozzle moving direction is compressive in nature, and three regions (I, II, and III) can be identified.

The first region (I) starts at coating-substrate interface up to about 11 µm thick, where the residual stress magnitude decreases from a peak value at the interface (x = 0) towards the coating surface. The high residual stress at the interface is due to the impact of the powders on the substrate surface, which was at room temperature initially. In this region, the temperature of the substrate and freshly deposited coating is not elevated significantly. Therefore, the thermal softening effect is not dominant. With the increase in coating deposition time, the heat generated by powder impacts, especially at the powder and substrate/coating interface, will diffuse into the substrate and the coating. Both the coating and substrate materials will be heated up subsequently, leading to increased thermal softening effects. Therefore, the residual stress magnitude starts to reduce as coating thickness builds up. The second region (II) is from 11 to 39 µm, where the residual stresses show a plateau. In this regime, the coating buildup process reaches a steady state, leading to stable heat generation and diffusion through the middle coating. The third region (III) is from 39 µm to the coating surface, where residual stress magnitude decreases to a minimum value. This is because fewer powders impact the newly formed coatings, and there is less kinetic energy input. Hence, the residual stress reaches the minimum level at the coating’s top surfaces.

The residual stresses in the whole coatings (black squares) and at the top 25 μm below the coatings’ surfaces (red circles) are averaged based on the data in Figure 10c for the four spray angles in order to illustrate the correlation of overall residual stress with spray angle (Figure 10d). The actual residual stress data measured from continuous Al2219 coatings at different traverse speeds and SoD in Figure 8d were averaged for three spray angles (70, 80, and 90°) and taken as reference (blue triangles) in Figure 10d. The simulation results of the averaged residual stress in the top 25 μm thick layer (which is close to the probing depth of Cr Kα X-ray along Al (311) plane by X-ray residual stress analysis) are comparable to the actual data measured at spray angles of 70°, 80°, and 90°, which validates the FEM computational model employed in this study. The compressive residual stress (CRS) clearly increased when the spray angle decreased (Figure 10d), which is attributed to the shear-dominated deformations of the powders caused by the tangential velocity (ν_t_) represented by νcosθ in Figure 8e, when the spray angles decreased from normal impact. This has been observed when we model the oblique impact of Al6061 powders [28] and Ti-6Al-4V powders [5] onto Al6061-T6 substrate via the finite element method (FEM). The oblique impacts of Al2219 powders are, therefore, expected to induce similar shear stress at the surface, which increases with a decrease in the spray angle from 90 to 70° (Figure 8c,d). The residual stress of the entire coating is much higher than the top 25 µm due to higher CRS close to the interface, as seen in Figure 8b.

The effect of spray angle can also be seen from the powders’ spatial and velocity distribution extracted from computational fluid dynamic (CFD) simulation (Figure 11). The y and z axis represent the spatial location coordinates in the lateral and vertical direction, respectively, while the impact velocity of the powders was indicated by the colormap. When the spray angle decreased from 90° to 75°, the cross-section of the particle plume from the nozzle exit still maintained a circular shape with a diameter of about 4 mm (Figure 11a,b). When the spray angle decreased further to 45°, a clear elongation of powder distribution in the vertical direction can be seen with vertical range of about 5 mm while the horizontal range was still about 4 mm. When the spray angle further decreased to 30°, the vertical distribution of the powders increased to 14 mm while the horizontal distribution still maintained at 4 mm. For spray angles of 45° and 30°, the vertical distribution of powders is clearly not centered but concentrates below the center point due to the inclination of the substrate. A change in spray angle does not change the powder velocity range significantly, which is in the range from 390 to 900 m/s for these four angles. The powders with high velocity (up to 900 m/s, yellow color) are typically concentrated in the center of the spot, while powders with low velocity (390 m/s, pink color) are present across the entire spot. As the decrease in spray angle (θ) led to a decrease in normal velocity (ν_n_) and an increase in tangential velocity (ν_t_, Figure 8e), a decrease in impact angle, therefore, led to a reduced ν_n_ at a smaller spray angle, a less deformed Al2219 powder with higher porosity (Figure 8c), much lower powder buildup but a wider deposit footprint (Figure 3a′). Therefore, the adhesion strength would be expected to decrease with spray angle with the concurrent increase of porosity, although adhesion tests of the coatings were not carried out due to the limited coating dimension and inconsistent thickness.

When spray angle and SoD are kept at the same level (i.e., 90° and 3 cm), CFD simulation shows that a lower traverse speed leads to a much longer footprint and denser powder streams impacting the substrate at the same unit time (Figure 12), when the spray gun has traveled about 90 cm from left to right at three traverse speeds. The lowest traverse speed (200 m/s, bottom panel) is calculated to have a much longer (in Y direction) and denser powder distribution compared to the one with medium traverse speed (350 mm/s, middle panel), which has a longer and denser deposit profile than the one with highest traverse speed (500 mm/s, top panel, Figure 12). The substrates need an “incubation” period before their surfaces are “activated” for powder deposition. Such an activation process is achieved by removing surface oxides by the first few impacting powders before subsequent powders can be deposited onto it. A low traverse speed thus provides more powders to activate the surface and then form the coating. More powders hitting the substrate would also lead to a higher heat buildup at low traverse speed. High substrate temperatures help to soften the substrate and preceding deposits. The combined effect led to a heavier weight increase (Figure 8a), a higher coating thickness (Figure 8b), and a reduced porosity (Figure 8c) observed in Al2219 after five passes when the traverse speed decreased from 500 to 350 mm/s and further reduced to 200 mm/s. These effects of traverse speed on Al2219 deposits agree largely with the observation when depositing Ti64 [18] and CoCr [19] powders at different traverse speeds. In addition, the lower buildup rate at high traverse speed (T3, 500 mm/s) led to a higher portion of rebounding particles which induced higher CRS at the substrate’s surface compared to that sprayed at the same angle by low traverse speed (T1, 200 mm/s, Figure 8d). If the traverse speed continues to increase above a critical or maximum value, it is foreseen that no deposition will occur [30], as the substrate cannot be activated by sufficient impact of powders.

Previous CFD simulation showed that bow shock’s thickness and width decreased with an increase in SoD [23]. Increasing SoD initially leads to an increase in deposition efficiency (DE), followed by an optimum stable middle region, before DE decreases with very long SoD [23]. A medium SoD range from 3 to 9 cm was thus selected in this work to avoid the low DE region in short and very long SoD. Indeed, at the high traverse speed of 500 mm/s, 3 cm SoD had a higher weight gain than 6 cm SoD (Figure 8a), which had a similar weight gain as that of 9 cm. Our CFD calculation results (Figure 13) show that SoD has no significant effects on the powder velocity, which is in a similar range from 390 to 900 m/s. However, a larger SoD leads to an increased spatial distribution which is more pronounced in the z direction (the injection direction) than in the y direction (the horizontal direction). More powders having low speed (represented by pink color) are spreading across much wider spots. At a fast traverse speed (T3, 500 m/s), the powder footprint is small, and the deposit’s density is low. A further widespread in the z direction at large SoD leads to a lower weight gain seen in Figure 8a, compared to smaller SoD at 70, 80, and 90° spray angles. However, at medium and low traverse speeds of 350 (T2) and 200 (T1) mm/s, the beneficial high powder flux impacting at substrate outweighs the impact of wider powder distribution in z direction arising from higher SoD. As a result, varying SoD from 3 to 9 cm does not lead to a noticeable difference in the weight gain among different angles when the gun was moved at a relatively low traverse speed.

With the understanding of the interplay between spray angle, traverse speed, and standoff distance on a single material system (i.e., Al2219), a continuous tool path movement can be planned in synergy with robot kinematics for the desired 3D geometry buildup, such as aluminum (and copper) stiffeners with sharp edges fabricated using a hybrid manufacturing system with two cooperative robotic arms: one for CS deposition and the other for machining and surface finishing [4]. In addition, CS has successfully produced near-net-shape square titanium frames with straight vertical walls and square corners using two cooperative robots [31]: One holding the CS gun while the other holding the build plate. These efforts demonstrate the feasibility of CS as a solid-state additive manufacturing technique for potentially a wide range of applications using various metallic powders.

## 5. Conclusions

In this work, the effects of spray angle, traverse speed, and standoff distance (SoD) were investigated holistically on one material system (Al2219 powders on Al2219-T6 substrate) using cold spray and verified with the simulations results by both finite element method (FEM) and computational fluid dynamic (CFD). For Al2219 powders, spray angle and traverse speed exercise have much more significant effects than SoD in determining deposits’ buildup. A decrease in spray angle from normal (90°) to oblique (θ < 90°) direction resulted in a lower normal velocity component (ν_n_ = ν_p_sinθ) of Al2219 powders. Hence, meaningful deposition can only occur at 80°~90° with much lower buildup at 70° and occasional single powder buildup at 60°. No deposition occurred at 50° and below for all the nine combinations under the three traverse speeds and three SoD. The critical angle was calculated by FEM to be between 55 and 60°, while CFD simulation shows that the powder footprint elongated from circular shapes at 90 and 75° spray angles to stretched-out ellipse shapes at 45 and 30°. All ellipses elongate along vertical direction. Indeed, shorter SoD (3 cm) was observed to result in higher coating buildup compared to longer SoD (6 and 9 cm) when the traverse speed is fast (i.e., ≥500 mm/s). Both experiments and CFD simulation have shown that low traverse speed benefits the coating buildup by having a high density of powders impacting the substrate at the same number of passes. Low traverse speeds (200~350 mm/s) overshadow the effects caused by SoD such that there was no significant difference when SoD was varied between 3 and 9 cm at low traverse speeds.

Combining high spray angle and low traverse speed led to high weight gain, thick coating with small porosity, and small compressive residual stress (CRS). Comprehending the effects of these three process parameters in a single Al2219 material system would assist the CS in repairing uneven surfaces and designing effective tool paths in cold spray additive manufacturing (CSAM) of 3D freeform structures using aluminum alloy powders.

## Figures and Tables

**Figure 1 materials-16-05240-f001:**
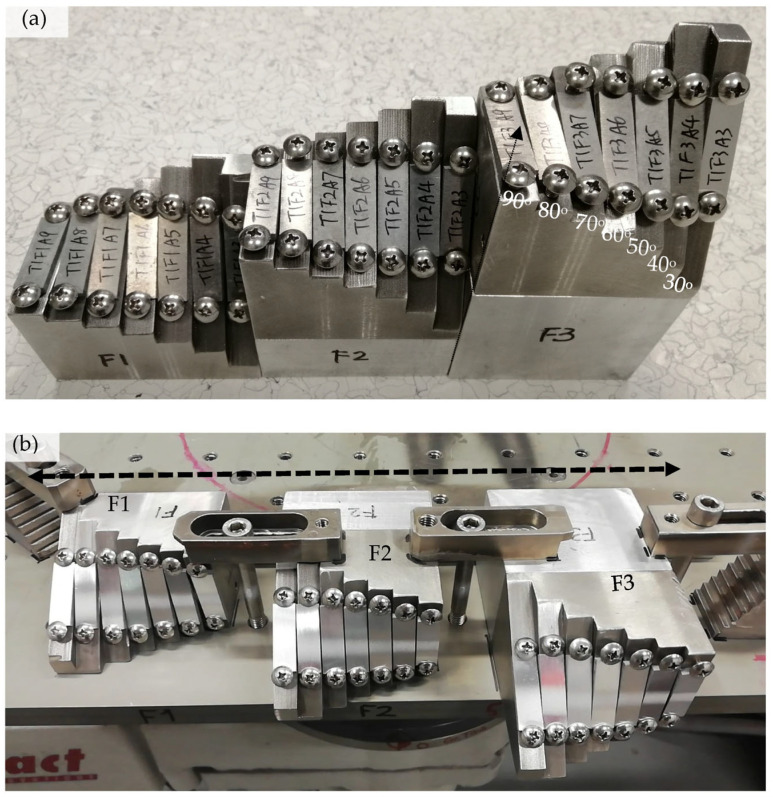
(**a**) Picture of three stainless steel fixtures with seven tilting angles from 90° to 30°. There were 3 cm and 6 cm thick aluminum spacers below the second fixture (F2) and the third fixture (F3), respectively. (**b**) Picture of 21 pieces of Al2219-T6 coupons clamped in the three fixtures after spraying five passes at a traverse speed of 500 mm/s. The bottoms of three fixtures were aligned while there was about 4 cm gap between each fixture.

**Figure 2 materials-16-05240-f002:**
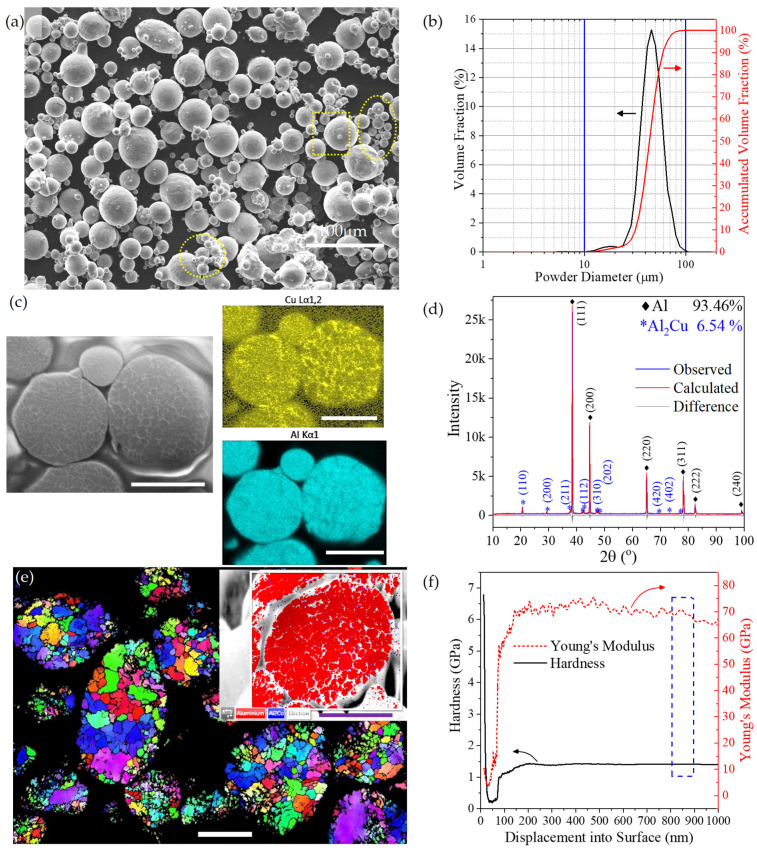
(**a**) SEM surface morphology as-received Al2219 powders. (**b**) Powder size distribution (PSD) after sonification in deionized water. (**c**) SEM image of the cross-section of the Al2219 powder and EDS elemental mapping of copper (Cu) and aluminum (Al) among the powder. (**d**) XRD analysis of Al2219 powders with Rietveld phase analysis results. (**e**) Inverse pole figure in z direction of Al2219 powders with inset at the top right corner showing phase map of one single powder with 0.2 µm step size. The scale bars in (**c**,**e**) represent 25 µm. (**f**) Hardness and Young’s modulus profiles vs. displacement depth into surface probed from the cross-section of Al2219 powder.

**Figure 3 materials-16-05240-f003:**
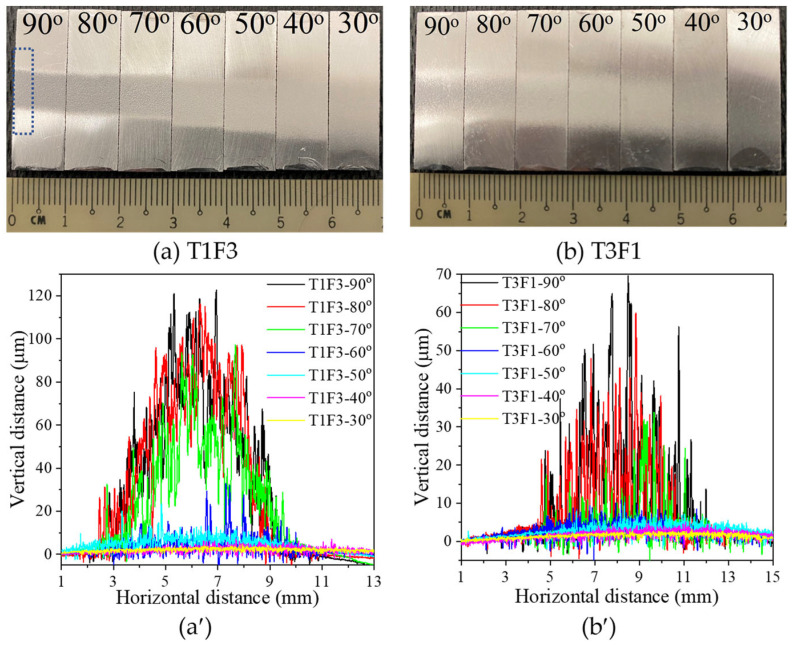
Pictures of two sets of Al2219 coupons deposited using (**a**) T1F3 and (**b**) T3F1 combination of parameters. Each set had seven coupons representing seven spray angles from 90 (far left) to 30° (far right). The respective contours of deposits measured by surface profilers are shown in (**a′**) and (**b′**), respectively. The dotted box in (**a**) represents a small piece cut from each coupon for subsequent tests.

**Figure 4 materials-16-05240-f004:**
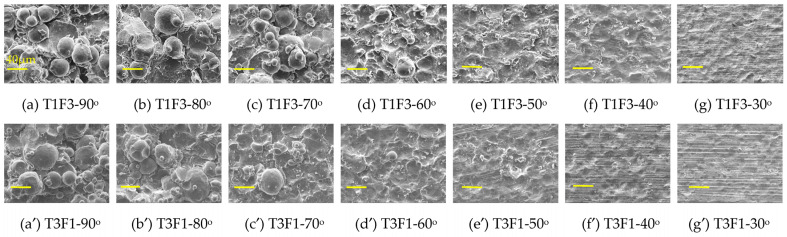
SEM surface morphology of the seven deposits in (**a**–**g**) T1F3 set and (**a′**–**g′**) T3F1 set, with each of them representing a spray angle from 90 (far left) to 30° (far right). The scale bar in each figure represents 40 μm.

**Figure 5 materials-16-05240-f005:**
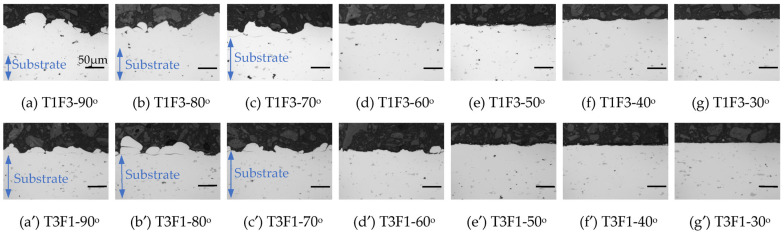
Optical microscope images of the cross-sections from the seven deposits in (**a**–**g**) T1F3 set and (**a′**–**g′**) T3F1 set of Al2219 deposits. Each picture represents a spray angle from 90 (far left) to 30° (far right). The scale bar in each figure corresponds to 50 μm. The occasional cracks between the coating and substrate serve as an indication of the interface.

**Figure 6 materials-16-05240-f006:**
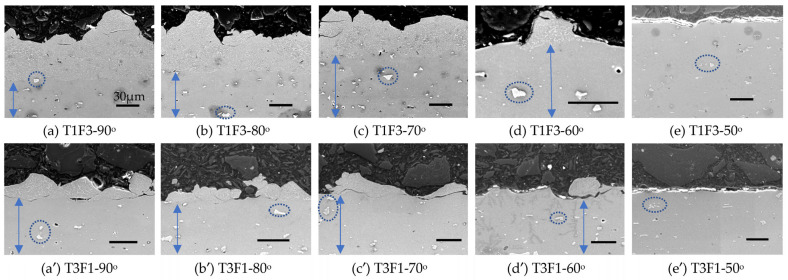
SEM images of the cross-sections from the five deposits representing a spray angle from 90 (far left) to 50° (far right) in (**a**–**e**) T1F3 set and (**a′**–**e′**) T3F1 set of Al2219 deposits. The scale bar in each figure represents 30 μm, and their lengths differ due to the different magnifications of each image. The images from 40 and 30° were similar to those at 50° and hence were not shown here. The blue arrows represent the Al2219-T9 substrate.

**Figure 7 materials-16-05240-f007:**
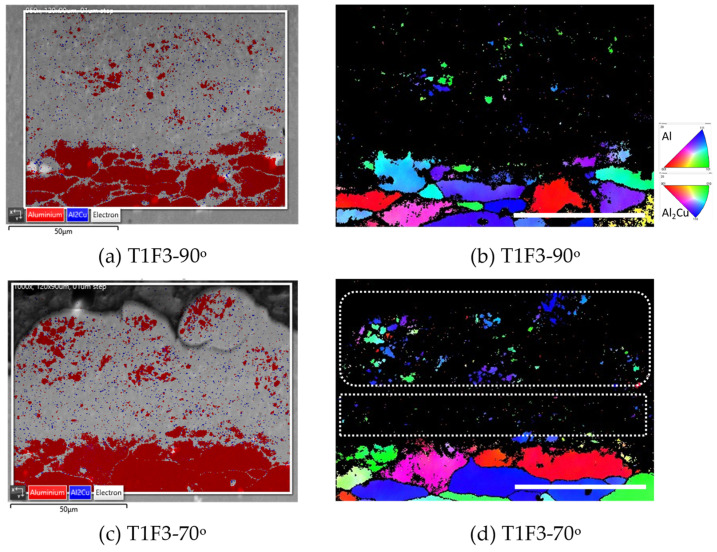
(**a**,**c**) SEM images overlay with phase maps and (**b**,**d**) inverse pole figures (IPF) in z-direction from electron backscattering diffraction (EBSD) analysis of the cross-section of Al2219 deposits sprayed at (**a**,**b**) 90 and (**c**,**d**) 70° spray angle in T1F3 set of samples. The white dotted boxes in (**a**,**c**) represented the areas for EBSD analysis in (**b**,**d**) with a step size of 0.1 µm. The scale bars in the four images represent 50 µm.

**Figure 8 materials-16-05240-f008:**
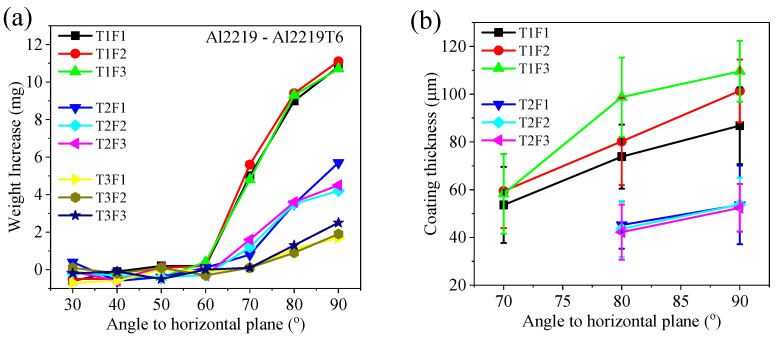
(**a**) Mass change, (**b**) coating thickness, (**c**) coating porosity, (**d**) residual stress of nine sets of Al2219 deposits along spray directions with respect to nine spray angles at the combination of three traverse speeds and three standoff distances. Only 15 Al2219 deposits with continuous coatings were selected to measure their thickness in (**b**) and porosity in (**c**). In (**d**), solid symbols represent continuous deposits, while open symbols represent scattered Al2219 deposits or shot-peened Al2219-T6 substrates. (**e**) shows the schematic decomposition of a powder’s velocity (ν) when it hits the substrate at an angle θ with respect to the surface plane.

**Figure 9 materials-16-05240-f009:**
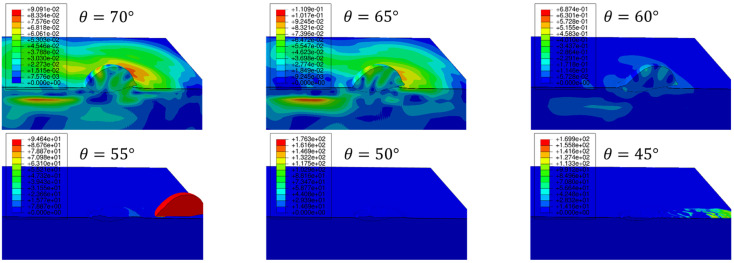
FEM simulation of single Al2219 powder’s velocity profile (in m/s) after impacting on Al2219-T6 substrate at an angle from 70 to 65, 60, 55, 50, and 45°.

**Figure 10 materials-16-05240-f010:**
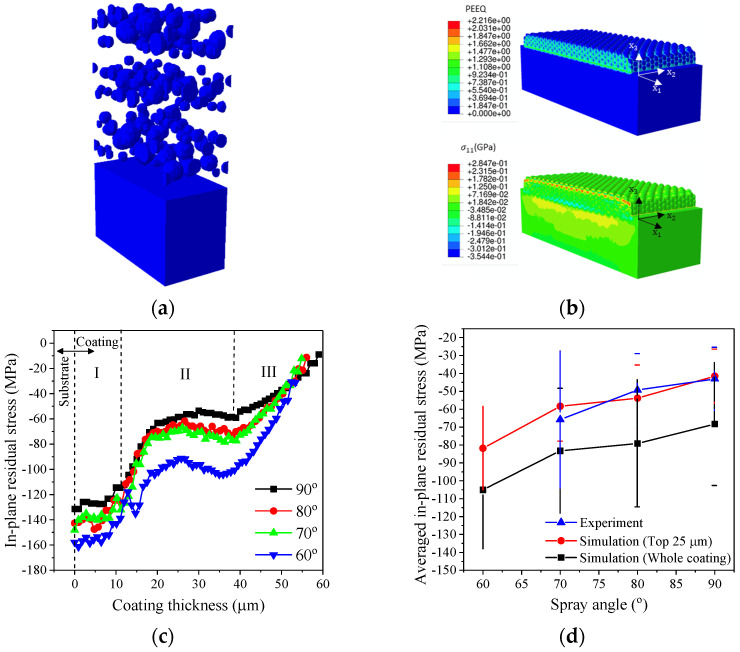
(**a**) Schematic diagram of multiple particle FEM computational model used for estimation of residual stresses in CS coating. (**b**) Simulation results of equivalent plastic strain (PEEQ) and von Mises stress after the deposition in 90° impact angle. (**c**) Simulation results of averaged in-plane residual stresses distribution (along spray direction) through the Al2219 coating thickness direction after the Al2219 coatings were deposited on Al2219-T6 substrate at four different spray angles. (**d**) Comparison of averaged residual stress results from (**c**) with reference experimental data based on Figure 8d by X-ray stress measurement.

**Figure 11 materials-16-05240-f011:**
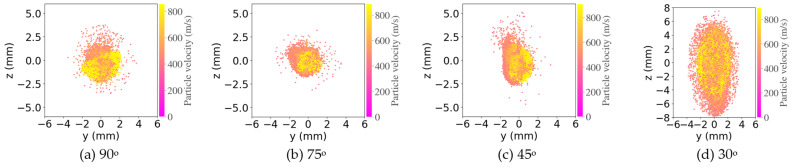
CFD simulation of Al2219 powders’ spatial distribution on Al2219-T6 substrate at 3 cm standoff distance with impact angles from 90 to 30°.

**Figure 12 materials-16-05240-f012:**
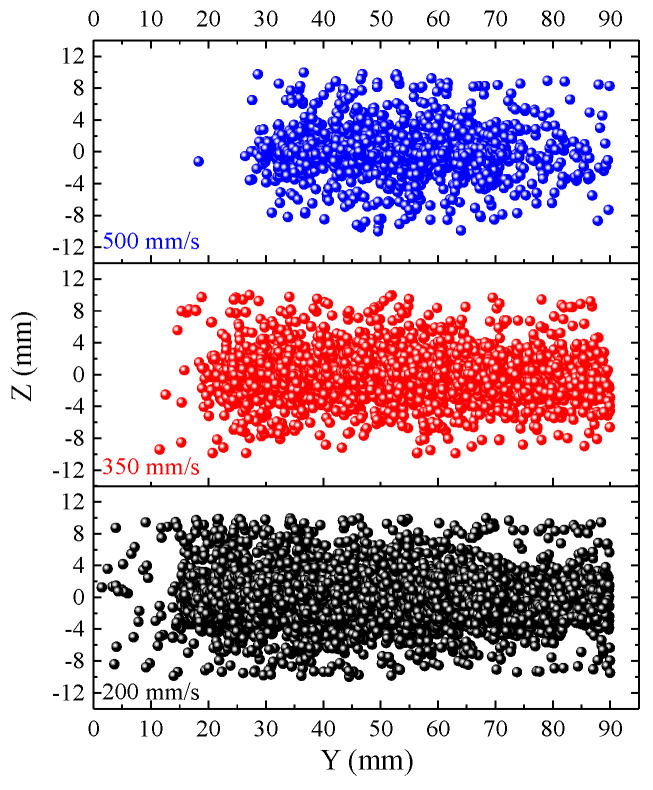
CFD simulation about the effects of three traverse speeds (500, 350, and 200 mm/s represented by blue, red and black dots, respectively) on powder spatial distribution at a SoD of 3 cm.

**Figure 13 materials-16-05240-f013:**
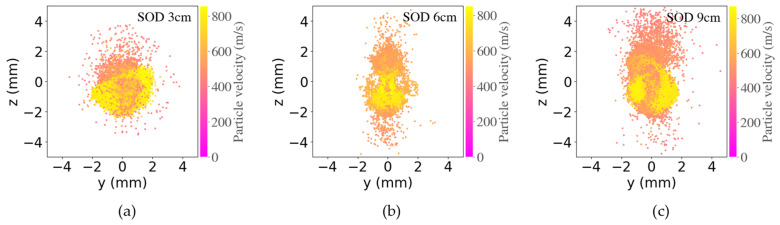
CFD simulation about the spot dimension (left vertical axis) and velocity distribution (right vertical axis) at three stand-off distances (SoD) of (**a**) 3, (**b**) 6, and (**c**) 9 cm.

**Table 1 materials-16-05240-t001:** The spray matrix at the combination of three traverse speeds and three standoff distances. There were seven angles at each condition.

Traverse Speed (mm/s)	N_2_ Pressure (MPa)	N_2_ Temperature (°C)	Pass	Standoff Distance (cm)		
				9 (F1)	6 (F2)	3 (F3)
200 (T1)	6	400	5	T1F1	T1F2	T1F3
350 (T2)				T2F1	T2F2	T2F3
500 (T3)				T3F1	T3F2	T3F3

**Table 2 materials-16-05240-t002:** Al2219 material properties used in FEM model [30].

Density (kg/m3)	2840	Young’s modulus (GPa)	73.1
A (MPa)	170	Reference strain rate (1/s)	1
B (MPa)	228	Specific heat (J/K kg)	0.864
*n* (dimensionless)	0.31	Thermal conductivity (W/m K)	116
*m* (dimensionless)	2.75	Melting temperature (K)	816
*C* (dimensionless)	0.028	Transition temperature (K)	298

**Table 3 materials-16-05240-t003:** Hardness and Young’s modulus of continuous Al2219 coatings at the combination of two traverse speeds, three spray angles, and three standoff distances (SoD). Yellow, pink and orange colors represent traverse speed, standoff distance and spray angle, respectively.

			Traverse Speed (mm/s)	**T1 (200)**			**T2 (350)**		
		Spray Angle (°)							
Hardness (GPa)		SoD (cm)		3 (F3)	6 (F2)	9 (F1)	3 (F3)	6 (F2)	9 (F1)
		90		1.69	1.71	1.69	1.56	1.56	1.65
		80		1.69	1.67	1.68	1.55	1.65	1.62
		70		1.70	1.73	1.67	-	-	-
Young’s modulus (GPa)		SoD (cm)		3 (F3)	6 (F2)	9 (F1)	3 (F3)	6 (F2)	9 (F1)
		90		81.50	78.50	76.50	72.10	73.20	77.00
		80		77.80	75.50	79.80	71.80	79.30	71.13
		70		75.00	78.40	74.70	-	-	-

## Data Availability

The raw/processed data required to reproduce these findings can be made available upon request.

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
