# Peer review of "Experimental and Numerical Study of Al2219 Powders Deposition on Al2219-T6 Substrate by Cold Spray: Effects of Spray Angle, Traverse Speed, and Standoff Distance"

_materials, 2023, doi:10.3390/ma16155240_

Round 1
Reviewer 1 Report
this is interesting paper with huge amount of experimental data, however, this data was not organized well. I suggest authors discuss the each parameters separately.
simulation result should be present in the result section not discussion.
1. authors should add the result of the F2 sample to the fig 3-6.
2. add scale bar to the SEM images
3.for figure 4 , it is better to compare the T1F1-.. with T1F2-.. and T1F3, in this case readers could easily find the effect of SoD and angles.
4. I believed the manuscript is not organized well , so first authors should reorganized the whole paper, after that i could put the specific comment.
authors tried to to investigated 3 parameters in same time, i could not say it is impossible but is hard. my suggestion, they have to study this parameters separately for example for SoD only focused on angle of 90, if they use the different angle the SoD for onside of sample to other side is changedAuthor Response
Please see the attachment

Reviewer 2 Report
The work investigates the effects of spray angle, traverse speed and standoff distance of AI2219 powders deposition by cold spray.
There are others papers that report the use of cold spray and others powders, however i didn´t find similar works using this technique and Al2219 over Al2219 substrate studying the reported in this paper, so i recommend to be published however it need some minor revisions:
1) It is necessary to separate images or figures or find a way to make them clearer to the reader.
2) Figure 2(d) need to be indexed (XRD)
3) Please provide little more details on how it was experimentally measured weight change, coating thickness, porosity and residual stress.
Reviewer 3 Report
In this work, the authors investigated the effect of spray angle, traverse speed, and standoff distance on the quality of cold spray AM. The experimental observations were supported by computational work using FEM and CFD models. However, the authors are recommended to address the following concerns before considering for publication.
· Why didn’t the authors try to capture an EBSD map at a higher magnification so as to get a better indexing?
· What is the methodology used to quantify the porosity in the deposits?
· What is the temper condition for AA2219 for which the JC model parameters were obtained (Ref. 30)? The powder and substrate are in different temper conditions. It is not sensible to use the same model parameters for both powder and the substrate. Assuming a yield strength of 170 MPa is an underestimate for T6 temper. The assumption will have its impact on all further calculations.
· What is the initial condition imparted to the powder particles? Is there any experimental evidence of particle velocity? Please list all the initial and boundary conditions. The details provided in the supplementary file need to be added to the main manuscript.
· Also, for multiple powder impact modeling, how did the authors define the contact between powders on their path to the substrate?
· What is the contact condition used for the contact interfaces? How did you model the interface? What contact condition is invoked for the powder to stick on to the substrate?
Minor comments
1. Take care of minor grammatical errors throughout the manuscript. For eg., ‘recent studies show’ in page 2.
2. Use et al. instead of et al (page 2)
3. Leave a space between the numerical value and unit, eg: 2 cm.
4. Avoid apostrophe in technical manuscript, Eg. change ‘powders’ morphology’ to ‘morphology of powder’.
5. Provide units for all the numeric values provided in the manuscript. Eg: 1.41 GPa and 68 GPa.
6. Typo 70° in page 11.

The authors need to improve and correct the grammatical and technical errors throughout the manuscript.
Round 2
Reviewer 3 Report
All comments are addressed properly, and the manuscript is recommended to publish.
